# Autophagosome Biogenesis

**DOI:** 10.3390/cells12040668

**Published:** 2023-02-20

**Authors:** Yan Zhen, Harald Stenmark

**Affiliations:** 1Centre for Cancer Cell Reprogramming, Faculty of Medicine, University of Oslo, Montebello, N-0379 Oslo, Norway; 2Department of Molecular Cell Biology, Institute for Cancer Research, Oslo University Hospital, Montebello, N-0379 Oslo, Norway

**Keywords:** autophagy, autophagosome, endosome, ESCRT, lipid transport, lysosome

## Abstract

Autophagy–the lysosomal degradation of cytoplasm–plays a central role in cellular homeostasis and protects cells from potentially harmful agents that may accumulate in the cytoplasm, including pathogens, protein aggregates, and dysfunctional organelles. This process is initiated by the formation of a phagophore membrane, which wraps around a portion of cytoplasm or cargo and closes to form a double-membrane autophagosome. Upon the fusion of the autophagosome with a lysosome, the sequestered material is degraded by lysosomal hydrolases in the resulting autolysosome. Several alternative membrane sources of autophagosomes have been proposed, including the plasma membrane, endosomes, mitochondria, endoplasmic reticulum, lipid droplets, hybrid organelles, and de novo synthesis. Here, we review recent progress in our understanding of how the autophagosome is formed and highlight the proposed role of vesicles that contain the lipid scramblase ATG9 as potential seeds for phagophore biogenesis. We also discuss how the phagophore is sealed by the action of the endosomal sorting complex required for transport (ESCRT) proteins.

## 1. Introduction

Macroautophagy (hereafter referred to as autophagy) is an evolutionarily conserved mechanism for the degradation of cytoplasmic material in lysosomes [1,2]. This process is activated under conditions of cellular stress or nutrient deprivation in order to metabolize non-essential macromolecules into amino acids and other small molecules needed for cellular homeostasis. Importantly, autophagy also serves as a mechanism for the degradation of cytoplasmic objects that can be toxic to the cell via a process known as selective autophagy. Such objects include damaged organelles, viruses, pathogens, and protein aggregates. Autophagy thus plays a major role both in cellular homeostasis and as a safeguard against hazardous bodies of endogenous and exogenous origin. The physiological importance of autophagy is underscored by multiple pathologies associated with its dysfunction, including cancers, neurodegenerative disorders, infectious diseases, myopathies, immunodeficiencies, and autoimmunity [2].

The autophagic process commences with the emergence of a sheet-like or cup-shaped membrane cisterna known as the phagophore or isolation membrane [3,4,5]. This structure expands and bends to sequester a portion of cytoplasm or a specific cytoplasmic object and eventually closes to form a double-membrane autophagosome. The autophagosome, in turn, fuses with an endosome to form an amphisome [6] or with a lysosome to form an autolysosome [3]. In amphisomes, and particularly in autolysosomes, the sequestered material is degraded by lysosomal hydrolases. Autolysosomes are eventually remodeled to reform lysosomes, which can engage in further rounds of autophagy [7].

Ever since its discovery, the phagophore has remained enigmatic. It has been estimated that about 3 million lipid molecules are required to form an autophagosome of 400 nm in diameter [8]. This raises the question of where the lipids come from and how the phagophore expands. Furthermore, how is the phagophore sculpted into a cup shape, and how does it close to forming an autophagosome? We will review some recent studies that have shed light on these issues, highlighting the potential sources of the phagophore membrane and the importance of lipid channels and scramblases.

## 2. The Preautophagosomal Structure as Autophagosome Generator in Budding Yeast

Light and electron microscopic studies of budding yeast exposed to nitrogen starvation have revealed a single preautophagosomal structure (PAS, also called a phagophore assembly site) located close to the vacuole, which is the yeast equivalent of lysosomes [9]. The PAS organizes a number of proteins involved in phagophore initiation and expansion (Figure 1 and Table 1). First, the protein kinase Atg1 and its complex partners Atg13, Atg17, Atg29, and Atg31 form the initial scaffold of the PAS, and subsequently, other Atg proteins, including a ubiquitin-like protein (Atg8), its conjugation machinery (Atg7, Atg3, Atg5-Atg12, Atg16), and a phosphatidylinositol 3-kinase (PI3K) complex (Vps15, Vps34, Atg6, Atg14) assemble onto this scaffold. Anchoring the PAS to the vacuole appears to be largely mediated by the interaction of the PAS protein Atg13 with the vacuole protein Vac8 [10].

The PAS has an amorphous appearance by electron microscopy, and recent results indicate that it is, in fact, a liquid-like condensate whose stability relies on the interactions of the disordered Atg13 proteins with Atg17 and other proteins [23]. The functional advantage of such a condensate is that a large number of proteins can be mobilized rapidly at a specific location. Under conditions of high nutrient availability, Atg13 becomes phosphorylated by the nutrient sensor complex TORC1, and this keeps it soluble in the cytosol. By contrast, upon starvation, Atg13 becomes dephosphorylated by PP2C phosphatases, and this results in its multivalent interactions with Atg17 and the resulting phase separation of the Atg1 complex [23,34]. Small (50 nm) Golgi-derived vesicles that contain the lipid scramblase Atg9 are recruited to the PAS, and this is followed by the recruitment of the PI3K complex, probably directly to the membranes provided by the Atg9 vesicles [35].

The PI3K complex is essential for starvation-induced autophagy [18], and this raises the question of why its catalytic product, phosphatidylinositol 3-phosphate (PI3P), is so important. One of the proteins that are recruited to the PAS is the PI3P binding protein Atg18, whose PI3P binding appears to be required for its localization to the PAS [32]. The elongating phagophore membrane indeed contains PI3P on its concave side [36]. Atg18 interacts with the phospholipid channel protein Atg2, which possibly tethers the PAS to the ER and, thereby, allows lipid transfer from ER during phagophore expansion [32]. The Atg2-Atg18 complex also binds to Atg9 on the PAS, thereby coordinating lipid transfer with transbilayer lipid scrambling [37]. Whether PI3P has additional functions during phagophore biogenesis other than directing the Atg18-Atg2 complex to the PAS is not entirely clear, but it is interesting to note that PI3P has to be dephosphorylated by the myotubularin-related phosphatase Ymr1 in order for autophagosomes to fuse with the vacuole [38]. This indicates that PI3P and its binding partner(s) must complete their function in phagophore biogenesis during a relatively narrow time window.

Central in autophagosome biogenesis is Atg8, whose C-terminal conjugation to phosphatidylethanolamine (PE) promotes its membrane binding and activity. This PE conjugation is mediated by a ubiquitination-like cascade involving the E1 activating enzyme Atg7, the E2 conjugating enzyme Atg3, and the E3 ligase complex consisting of Atg16 and the Atg5-Atg12 conjugate (which itself requires Atg7 and the E2 enzyme Atg10) [39]. The exact functions of PE-conjugated Atg8 in phagophore expansion still remain to be clarified. In vitro reconstitution approaches have indicated several biochemical activities of Atg8-PE, including vesicle tethering, hemifusion, and fusion [40]. Promoting the fusion of the phagophore with Atg8-PE-containing vesicles would be a potential way to induce its expansion, but it has also been shown that the insertion of Atg8-PE into membrane vesicles increases the area difference between the outer and inner membrane leaflet, which promotes the formation of tubulovesicular structures that might cause phagophore expansion [41].

## 3. Similarities and Differences between Budding Yeast and Mammals in Autophagosome Biogenesis

The remarkable conservation of the autophagic machinery components between yeast and mammals indicates that the basic principles of autophagosome biogenesis have been conserved through evolution. However, whereas the PAS has a central role in autophagosome biogenesis in yeast, no direct equivalent to the PAS has so far been identified in mammalian cells. This does not necessarily mean that no such equivalent exists since discrete phase-separated PAS-like structures would be difficult to detect in the crowded cytoplasm of mammalian cells, especially if the structures are short-lived. However, it is also possible that mammalian ATG proteins could assemble on membranes in an organized pattern instead of forming liquid-like assemblies.

Mammalian cells are generally much larger than yeast cells and typically have hundreds of lysosomes compared with single vacuoles in yeast cells. Perhaps owing to these differences, multiple membranes have been identified as potential sources of autophagosome membranes in mammals, and these are discussed in the following sections.

## 4. The Endoplasmic Reticulum as Source of Phagophores

In most mammalian cells, the endoplasmic reticulum (ER) extends as a tubulocisternal network throughout the cell body. It is thus well positioned as a potential source for autophagic membranes, and there is indeed good evidence that the ER plays an important role in autophagosome biogenesis. Electron tomography shows that phagophores are often tightly associated with ER membranes, either in continuity or closely apposed [42,43]. In fact, quantitative 3D light and electron microscopy indicate that all phagophores are associated with the ER [44]. The PI3P-binding ER protein DFCP1/ZFYVE1 is frequently observed at sites of phagophore biogenesis during starvation. DFCP1-positive membrane structures typically have an omega shape during starvation-induced autophagy and are, therefore, referred to as omegasomes [27]. Even though DFCP1 is nonessential for bulk autophagy, omegasomes have been proposed to form domains of the ER that function as cradles for nascent phagophores. There is some disambiguity in the literature on whether the ER membranes are continuous with the forming phagophore or not [42,43,45]. Although both scenarios may be correct, a physical separation between the ER and the growing phagophore would be most consistent with what is known about phagophore assembly in the yeast and also with recent studies that involve ATG9-containing vesicles.

The discoveries that ATG9 is a lipid scramblase [21,22] and ATG2 a lipid channel [12] have offered a plausible mechanism for how the ER can provide lipids to the growing phagophore. Single or multiple ATG9 vesicles coming from the Golgi could act as the seed for phagophore formation [46]. Upon docking to the ER via ATG2, lipids synthesized in the ER could be transferred to the ATG9-containing phagophore seed, thereby promoting its expansion. ATG9, which forms a complex with ATG2 [47], would function to equilibrate the channeled lipids into the inner phagophore membrane bilayer in order to allow membrane expansion. On the ER side, two other lipid scramblases, VMP1 and TMEM41B, would have an equivalent role in equilibrating levels of newly synthesized lipids over the ER membrane. ATG2 attaches to the phagophore membrane via the binding of its C-terminus to ATG9 and the Atg18 homolog WIPI4, whereas its N-terminus interacts with VMP1 and TMEM41B in the ER membrane [48] (Figure 2). In addition to ATG2, the structurally related VPS13 proteins have also been implicated in autophagy [49,50,51], indicating a role for multiple lipid channels in phagophore expansion.

This model of phagophore expansion via lipid transfer from the ER is attractive because it provides a rationale for the importance of ATG9-containing vesicles, whose function has previously remained enigmatic, and also offers possible explanations for the double membrane morphology of the phagophore and the observation that the outer membrane of the phagophore is largely devoid of transmembrane proteins [52]. Nevertheless, it does not exclude the possibility that phagophores can form directly from organelle membranes, as discussed below. The expansion of the phagophore through its fusion with small vesicles that contain ATG9 or Atg8 proteins is also plausible. It is worth noting that the recycling of ATG9 from endosomes to the trans-Golgi network is required to sustain autophagy, presumably in order to avoid the exhaustion of the ATG9 pool during autophagy [53].

## 5. COPII Vesicles and ER Exit Sites in Phagophore Biogenesis

ER exit sites are specialized regions of the ER where COPII transport vesicles are generated. These vesicles, which are best known for their role in ER-Golgi transport [54], are interesting in the context of autophagy since COPII subunits are essential for starvation-induced autophagy in budding yeast [55]. In this organism, autophagosomes are formed very close to ER exit sites, and the function of these sites is required for PAS formation [56]. Under starvation conditions, the secretory pathway is largely shut down, and COPII vesicles, instead of fusing with the Golgi, are diverted to the PAS. If autophagy is blocked, the COPII vesicles accumulate at the PAS via binding to Sec23. The small GTPase Ypt1 (homolog of mammalian RAB1) is activated by its guanine nucleotide exchange factor complex, TRAPPIII, and, together with Atg17, these proteins serve to tether Atg9 vesicles to COPII vesicles [57], thereby providing potential seeds for the phagophore.

Although most information on COPII and ER exit sites in autophagy has been obtained through studies of budding yeast, there is evidence that similar mechanisms could operate in mammalian cells. In particular, a mammalian TRAPPIII complex has been found to function as a GEF for RAB1 and to regulate autophagy, though it appears to regulate the trafficking of ATG9 vesicles from recycling endosomes to the Golgi instead of tethering ER-derived vesicles to the forming phagophore [58,59]. In addition to its proposed role in vesicle tethering, RAB1 has been shown to interact with the autophagy-specific PI3K complex [60], and yeast Ypt1 interacts with the autophagy-initiating kinase Atg1 at the PAS [61]. This raises the possibility that RAB1, trafficked on COPII vesicles, might play a role in Atg1/ULK1 and PI3K recruitment during the initiation of autophagy.

## 6. Mitochondria as Source of Phagophore Membranes

As major sites for ATP production and lipid metabolism, mitochondria are central to cellular metabolism. Mitochondrial membranes, which consist of two layers, are highly dynamic and undergo fusion, fission, and budding reactions that are controlled by metabolic cues. An interesting aspect of the outer mitochondrial membranes is that they have been implicated as membrane sources for starvation-induced autophagosomes [62]. Evidence for this includes the detection of ATG5 on mitochondria during starvation, the translocation of a targeting sequence derived from a mitochondrial outer membrane protein to autophagosomes, and the delivery of fluorescently tagged lipids from mitochondria to autophagosomes. Moreover, the depletion of Mitofusin2, which not only mediates mitochondrial fusion but also forms contact sites between the mitochondria and ER, strongly inhibits starvation-induced autophagy.

Analyzing the involvement of mitochondria in autophagosome biogenesis is complicated by the fact that damaged mitochondria, which might occur during starvation, are themselves targeted by autophagy in a process known as mitophagy [63]. However, under starvation conditions that are sufficient to induce translocation of a mitochondrial outer membrane reporter to autophagosomes, no mitochondrial matrix proteins could be detected in the autophagosomes, arguing that the detected contribution of mitochondrial membranes to autophagosomes is not due to mitophagy [62].

How well suited is the mitochondrial outer membrane as a source of phagophore membranes? This membrane contains abundant mitochondrial outer membrane proteins, and these are generally excluded from autophagosomes, suggesting that there must exist some gating mechanism that prevents most mitochondrial proteins from entering phagophores [62]. On the other hand, it is interesting that PE, which is conjugated to Atg8 family proteins to mediate their membrane anchoring and, thereby, their function in phagophore expansion, is synthesized in both mitochondria and the ER [64]. Mitochondrial outer membranes could, therefore, serve as a source of PE.

## 7. Contact Sites between ER and Mitochondria as Source of Phagophores

A model that unifies the notions of phagophore biogenesis from ER and mitochondria, respectively, has emerged through the observation that the membrane-targeting component of the autophagic PI3K complex, ATG14, is recruited to contact sites between ER and mitochondria upon starvation, by binding the ER-resident SNARE protein Syntaxin-17 [65]. The depletion of PACS-2, a protein important for the formation of ER-mitochondria contacts [66], strongly inhibits the recruitment of ATG14 and DFCP1 to ER-mitochondria contact sites during starvation and also decreases the PE conjugation of the Atg8 family protein LC3, indicating that ER-mitochondria contacts are involved in phagophore biogenesis [65]. The finding that ATG9-containing vesicles are mobilized to these sites is consistent with this idea [67]. It still remains to be understood how ATG14 and DFCP1 are targeted specifically to ER-mitochondria contact sites by Syntaxin-17 during starvation and exactly how the phagophore forms at the contact sites.

## 8. Endosomes as Source of Phagophore Membranes

ATG9 is the only transmembrane protein of the core autophagic machinery, and its delivery to the growing phagophore depends on vesicle transport. ATG9 resides in the Golgi under conditions of low autophagy, whereas it is mobilized to sites of phagophore growth during starvation. This mobilization occurs via recycling endosomes that contain ATG16L1 and the small GTPase RAB11. The PX-domain-containing protein SNX18 is important for the budding of ATG9- and ATG16L-containing vesicles from recycling endosomes which, in concert with the large GTPase Dynamin 2, is thought to mediate the scission of vesicles from the endosome membrane [68,69]. A number of other endosomal SNX family members have also been implicated in autophagy, including SNX1, SNX2, SNX4, and SNX5 [53,70,71]. This underscores the importance of vesicle budding from endosomal tubules to sustain autophagosome biogenesis.

While recycling endosomes have primarily been regarded as a source of ATG9 vesicles in transit to sites of phagophore biogenesis, they have also been proposed to function directly as platforms for phagophore assembly [72]. The PI3P binding Atg18-related protein WIPI2 has been found to bind to RAB11A on recycling endosomes which co-localize with autophagosome markers, and the loss of RAB11A inhibits the recruitment of autophagy components and prevents starvation-induced autophagy. However, direct evidence that phagophores form from recycling endosomes has not been presented.

## 9. The Plasma Membrane as Source of Phagophore Membranes

ATG16L1 is central in phagophore biogenesis because it functions as part of an E3 enzyme for the PE conjugation of Atg8 family proteins. This protein is associated with membranes and thereby contributes to the membrane targeting of the autophagy machinery. One of the interactors of ATG16L1 is the clathrin-heavy chain, and inhibition of clathrin-mediated endocytosis inhibits both the starvation-induced formation of ATG16L1-containing puncta and the biogenesis of autophagosomes [73]. Therefore, the plasma membrane has been proposed to act as a membrane reservoir for phagophores during conditions of increased autophagy. Another possibility is that ATG16L1-containing endosomes, resulting from endocytosis, are important for the delivery of ATG16L1 vesicles to the phagophore (see below).

## 10. Lipid Droplets as Source of Phagophore Membranes

Lipid droplets, as cellular vehicles for the storage of neutral lipids, grow from the ER by mechanisms that use some of the same components as phagophores [74]. This has raised the question of whether lipid droplets contribute lipids to phagophores. In budding yeast, the deletion of enzymes responsible for triacylglycerol or steryl ester biosynthesis results in the inhibition of starvation-induced autophagy, and this is also the case with the deletion of proteins that form contact sites between lipid droplets and the ER [75]. Likewise, in mammalian cells, a neutral lipase that localizes to lipid droplets, PNPLA5, appears to be needed for the optimal initiation of autophagy [76]. Collectively, these findings suggest that lipid droplets could indeed play a role in autophagy.

However, there is also the possibility that interference with lipid droplet biology could affect autophagy indirectly, and it has been reported that lipid droplets as such are dispensable as a source for autophagosomes in budding yeast, whereas their absence causes alterations to the ER due to imbalanced phospholipid composition and ER stress which indirectly inhibit phagophore biogenesis [77].

## 11. Hybrid Golgi-Endosome Membranes as Source of Phagophores

Although a direct equivalent to the yeast PAS has not been identified so far in mammalian cells, a structure with PAS-like properties, termed hybrid PAS (HyPAS), has recently been described [78]. The HyPAS is formed through the fusion of cis-Golgi-derived vesicles that contain the ULK1 complex partner FIP200 with endosome-derived vesicles that contain the E3 protein ATG16L1. Fusion requires the SNARE proteins Syntaxin-17 on Golgi-derived membranes and VAMP7 on endosome-derived membranes and is stimulated by Ca^2+^ via two Ca^2+^ binding Syntaxin-17 interactors: E-SYT2 and SIGMAR1. At the same time, Syntaxin-17 interacts with the ER Ca^2+^ pump protein SERCA2 to inhibit its function and, thereby, contribute to elevated Ca^2+^ levels [78].

Both COPII vesicles and vesicles derived from the plasma membrane or endosomes have been implicated in autophagosome biogenesis (see above), and the existence of the HyPAS could potentially explain the observations that both vesicles from the biosynthetic and endocytic pathways contribute to autophagosome biogenesis. The exact involvement of the HyPAS in phagophore biogenesis still needs to be defined and it is important to establish whether it serves as a direct source of the phagophore membrane or whether it organizes the ATG machinery in a similar fashion as the yeast PAS.

## 12. Closure of the Phagophore

When phagophore expansion is complete, the cup-shaped phagophore needs to close in order to form an intact autophagosome. Such a closure of the hole in a double membrane is topologically equivalent to the inward budding of a vesicle in a single membrane [79], as occurs during the biogenesis of multivesicular endosomes [80]. The formation of intraluminal vesicles (ILVs) in endosomes is mediated by the endosomal sorting complex required for transport (ESCRT): a complicated molecular machinery that consists of three subcomplexes, ESCRT-I, -II, and –III, and several accessory components, including the ATPase VPS4 [28]. Studies of both bulk and selective autophagy have shown that the ESCRT machinery also serves to seal the phagophore [25,26,30,81]. The ESCRT-I subunit VPS37A localizes to the phagophore via an N-terminal domain that is unique to this isoform and not found in the other VPS37 isoforms, VPS37B, VPS37C, and VPS37D. The VPS37A-containing ESCRT-I variant recruits ESCRT-III, which is thought to assemble into multimeric filaments that physically mediate membrane closure in concert with the ATPase VPS4 (Figure 2). Interference with ESCRT-I, ESCRT-III, or VPS4 functions prevents phagophore closure.

Even though VPS37A seems to direct the ESCRT machinery to seal the phagophore, it remains to be established how this ESCRT-I subunit is targeted to the hole that needs to be closed. Studies from budding yeast have implicated the small GTPase Vps21 (homolog of RAB5) upstream of ESCRT recruitment. Vps21 recruits the Atg1 complex subunit Atg17 (homolog of mammalian FIP200), which can interact directly with the ESCRT-III subunit Snf7 (corresponding to mammalian CHMP4) [81]. However, as studies in mammalian cells have shown that ESCRT-I is upstream of ESCRT-III recruitment [25,30], further studies are required in order to understand the contributions of Vps21/RAB5 and Atg17/FIP200 in ESCRT recruitment to the closing phagophore.

Although ESCRTs mediate the final closure of phagophores, several other proteins contribute to closure because they are involved in phagophore growth and shaping, which are essential prerequisites for eventual closure. For instance, members of the Atg8 family of ubiquitin-like proteins, which mediate phagophore expansion, are required for phagophore closure [82], and so are some of their regulators [83,84,85]. 

Whenever phagophore closure fails, the degradation of autophagocytosed cargo, known as autophagic flux, is inhibited [25,26,30,81]. This can probably be attributed to a failure in lysosomes to fuse with unsealed phagophores, as ESCRT inactivation is known to cause an accumulation of “autophagosomes” (presumably representing unsealed phagophores with only a small opening) that do not progress to autolysosomes [86,87,88,89]. The SNARE protein Syntaxin-17, which mediates the autophagosome-lysosome fusion upon forming a SNARE complex with the lysosomal SNAREs SNAP29, and VAMP7/8, has been reported to associate selectively and transiently with sealed autophagosomes [90,91], and this provides a plausible explanation for the failure of unsealed phagophores to fuse with lysosomes. However, the mechanistic basis for the recruitment of Syntaxin-17 to sealed autophagosomes only has not yet been determined.

## 13. Conclusions: Challenges and Perspectives

In spite of decades of intensive research on autophagosome biogenesis, we still do not have a complete picture of how the autophagosome forms and shapes and where its membrane constituents come from. Nevertheless, the recent convergence of genetic, cell biological, biochemical, and biophysical studies has provided important progress that makes it possible to propose rational models for autophagosome biogenesis. In particular, the realization that ATG2 is a lipid channel and ATG9 is a lipid scramblase has led to new insights into how the phagophore can grow by receiving newly synthesized lipids [46]. Since the ER is the main site of lipid biosynthesis, this also consolidates the view that phagophores mainly originate from the ER: a concept that is supported by 3D electron microscopy [44]. Assuming that the phagophore at least in part grows by receiving lipids transferred from the ER, this nevertheless raises the question of whether and how lipid biosynthesis is coupled to phagophore growth.

The key role of the ER in phagophore biogenesis does not by any means imply that other membranes are unimportant. Mitochondria also contribute to lipid biosynthesis, and they form abundant contacts with the ER that are likely the origins of phagophore biogenesis [65]. ATG9-containing vesicles are thought to constitute seed membranes for phagophores, and the trafficking of ATG9 to sites of phagophore biogenesis necessarily has to involve multiple membrane compartments since this polytopic membrane protein is synthesized in the rough ER and can follow several alternative transport routes after it exits the ER [92]. ATG9 transits through COPII vesicles (or vesicles piggybacking on those) as well as the plasma membrane and endosomes, and this might, in part, explain the importance of such membranes for phagophore biogenesis (Figure 3). The E3 enzyme ATG16L1 is also membrane-bound, although this protein contains no transmembrane domain but associates with membranes through an N-terminal amphipathic helix and a C-terminal segment [93]. Similar to ATG9, ATG16L can be found on multiple membranes, including clathrin-coated vesicles budding from the plasma membrane and recycling endosomes [72,73]. ATG16L1 trafficking might, therefore, account for some of the involvement of such membranes in phagophore biogenesis. This is also the case with the ULK1 complex protein FIP200, which redistributes to phagophores during starvation [94]. Indeed, the fusion of ATG16L1-containing vesicles originating from endosomes with FIP200-containing vesicles from the Golgi has been found to result in a hybrid compartment with PAS-like properties [78].

Given the importance of Atg8 family proteins in phagophore growth, and the numerous studies on these proteins, it is a paradox that we still do not understand exactly how these proteins function in phagophore expansion. Atg8 proteins are not strictly essential for autophagosome formation, but autophagosomes are much smaller in their absence [95]. Furthermore, Atg8 proteins are not specific for phagophores and autophagosomes but can also label single membranes (mainly conjugated with phosphatidylserine instead of PE) [96] and can even be conjugated with proteins [24]. The in vitro fusogenic activity of yeast Atg8 has spurred the notion that Atg8 proteins could contribute membranes to the growing phagophore through vesicle fusion [40], and there is also evidence that Atg8 proteins promote autophagosome-lysosome fusion [95,97]. There are six members of the human Atg8 family, and different family members are likely to play differential roles in phagophore expansion and autophagosome-lysosome fusion [98]. A large number of proteins can interact with Atg8 proteins through LC3-interacting regions (LIRs) that bind to the LIR docking site (LDS) in the Atg8 protein [20], and incorporation of cargoes for selective autophagy is likely to be a key function for Atg8 proteins. Such cargo sequestration may, itself, play a role in directing phagophore growth. Through interaction with the autophagy-initiating ULK kinase complex, Atg8 proteins might also provide positive feedback for phagophore initiation [99]. In any case, understanding how Atg8 proteins help make the phagophore and autophagosome remains a continuous challenge in autophagy research.

Another challenge is to understand how the phagophore is sculpted to achieve a cup-shaped double membrane. How is the distance between the two membranes kept during growth and bending, and which forces cause the bending? An understanding of this requires a combination of biophysical, biochemical, and genetic studies, perhaps aided by mathematical modeling. Curvature-sensing and -promoting proteins are likely to play important roles in phagophore sculpting, and several such proteins are known to be involved in autophagy, including the autophagy-specific PI3K subunit ATG14 [100], the PI3K-associated protein BIF-1 [101], and the E3 enzyme ATG16L1 [102]. It is possible that DFCP1-containing ER domains, omegasomes, may help in shaping the phagophore from the outside [27]. At the same time, there is evidence that a scaffold may help sculpt the phagophore from the inside. The actin capping protein CapZβ binds to PtdIns3P on the phagophore to stimulate actin polymerization and the formation of an actin scaffold inside the phagophore [103]. Further work is required in order to understand how the actin scaffold is formed selectively on the inside and how its dynamics are regulated to achieve the bending of the phagophore.

It seems clear that the closure of the phagophore is mediated by ESCRT-III, recruited via a VPS37A-containing version of ESCRT-I. The mechanism of ESCRT-III recruitment by ESCRT-I in the context of phagophore closure has not been investigated in detail, but from other ESCRT activities, it is known that the ESCRT-I subunit TSG101 can interact directly with the ESCRT-III subunit CHMP4. There are also accessory components that can bridge ESCRT-I and ESCRT-III [28]. However, an outstanding question is how VPS37A and the rest of ESCRT-I are recruited specifically for the opening in the phagophore. This opening is characterized by a very high membrane curvature, and it is conceivable that ESCRT recruitment involves a structural element that binds specifically to highly curved membranes. It is important to identify such a structure.

Our recent insight into how the autophagosome is formed is also valuable outside the field of autophagy since autophagy mechanistically intersects with many other cellular processes and plays profound roles in both the prevention and progression of various human diseases. With a better mechanistic understanding of autophagosome biogenesis, we are gaining more targets for intervention with autophagy for future disease management [2].

## Figures and Tables

**Figure 1 cells-12-00668-f001:**
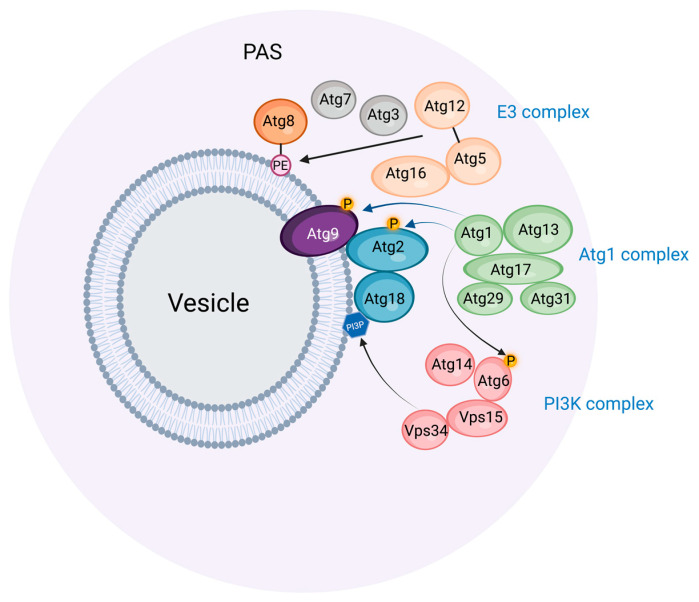
The preautophagosomal structure (PAS) in budding yeast. The PAS is thought to be a phase-separated molecular assembly, which associates with Atg9-containing vesicles. The Atg1 complex provides the scaffold for the PAS, and catalyzes the phosphorylation of Atg2, Atg9, and Atg6. The PI3K complex causes the formation of phosphatidylinositol 3-phosphate (PI3P) in membranes associated with the PAS. The E3 complex mediates the conjugation of Atg8 with phosphatidylethanolamine (PE) in conjunction with the E1 enzyme Atg7 and the E2 enzyme Atg3. Prepared with BioRender (www.biorender.com) and accessed on 15 February 2023.

**Figure 2 cells-12-00668-f002:**
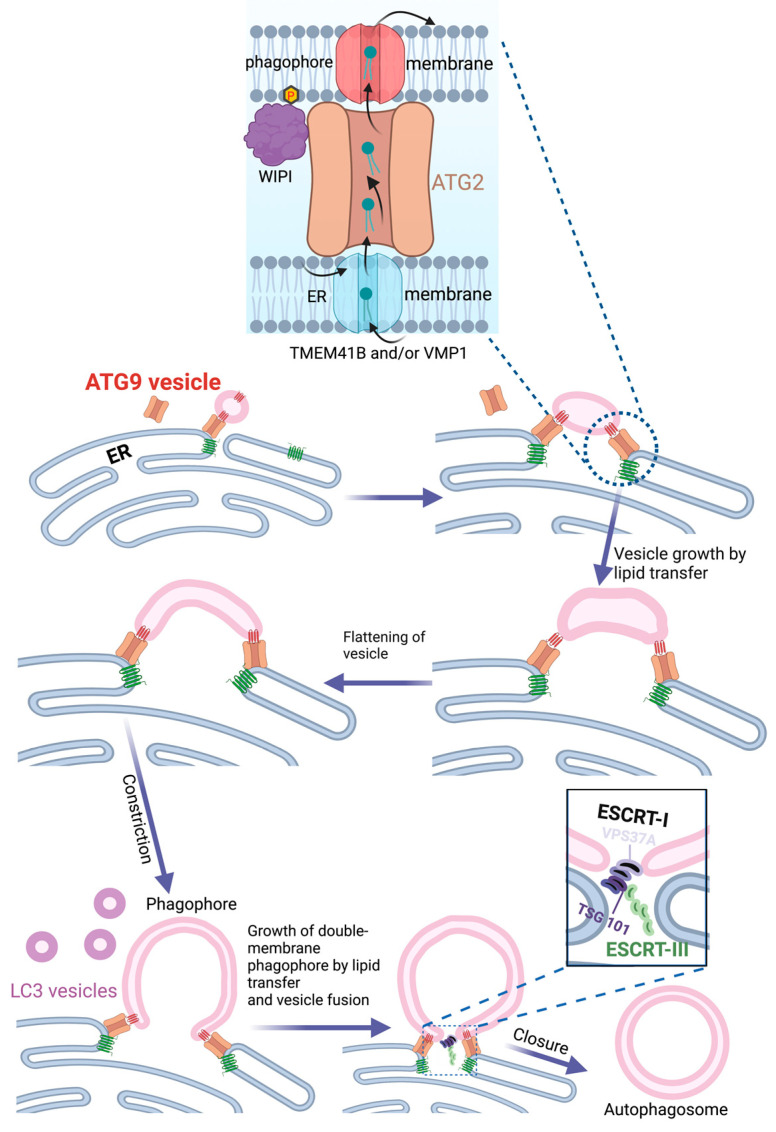
Lipid transfer from ER during phagophore biogenesis. The lipid channel ATG2 interacts with the lipid scramblases TMEM41B and VMP1 in the ER and ATG9 on the phagophore seed vesicle. Newly synthesized phospholipids in the ER are transferred from the luminal to the cytosolic side, where they are channeled by ATG2 to the seed vesicle. Transbilayer lipid transfer by ATG9 ensures phagophore growth. The model illustrates how the seed vesicle transforms into a double-membrane phagophore by flattening, growth, and bending. A variant of ESCRT-I that contains VPS37A is recruited to the remaining hole in the phagophore via the N-terminus of VPS37A. This causes the recruitment and activation of ESCRT-III, presumably via interactions between TSG101 in ESCRT-I and the ESCRT-III subunit CHMP4B. ESCRT-III forms filaments that close the phagophore to form an autophagosome. Prepared with BioRender (www.biorender.com) and accessed on 15 February 2023.

**Figure 3 cells-12-00668-f003:**
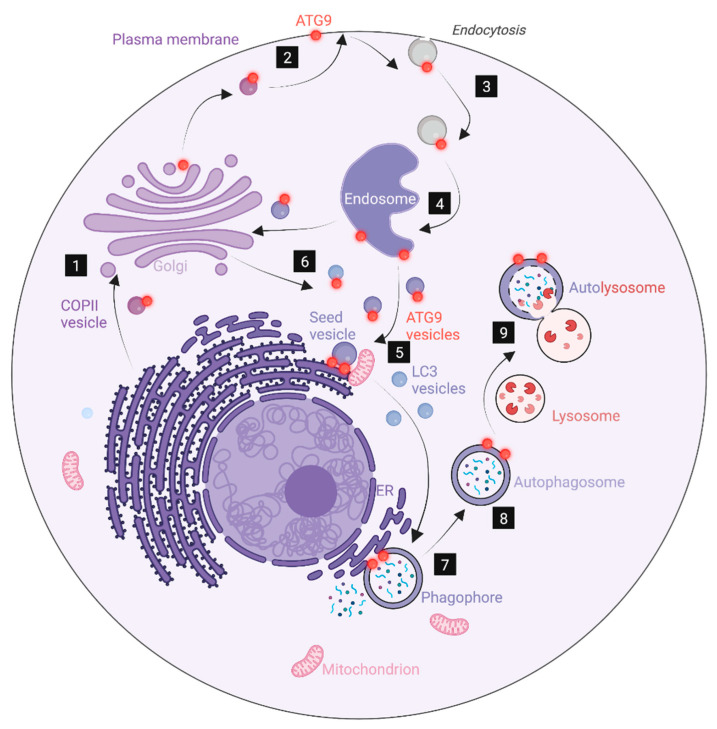
An ATG9-centric model for autophagosome biogenesis. ATG9 molecules (red) follow the biosynthetic pathway from the ER (1) to the plasma membrane (2) via the Golgi. ATG9 at the plasma membrane is endocytosed (3) and reaches endosomes (4), from where it can enter into vesicles that fuse to form seed vesicles for phagophores (5), possibly together with ATG9-containing vesicles coming from the biosynthetic pathway (6). Lipids generated in the ER and mitochondria are transported from the ER to the seed vesicle via the lipid channels ATG2 and VPS13 (not indicated). In conjunction with ATG9-mediated transbilayer lipid transfer, this ensures vesicle growth, possibly aided by fusion with vesicles that contain Atg8 proteins. Upon growth, the vesicle flattens to form the phagophore (7), which ultimately closes to form the autophagosome (8). The autophagosome fuses with a lysosome to form an autolysosome (9), in which the engulfed cytoplasm is degraded. In this model, the contributions to autophagy of ER-Golgi transport, endocytosis, and endosomal budding are related to their involvement in ATG9 trafficking. Prepared with BioRender (www.biorender.com) and accessed on 15 February 2023.

**Table 1 cells-12-00668-t001:** Proteins involved in autophagosome biogenesis. For *H.sapiens* proteins, isoforms have mostly been omitted from the table.

Protein		Function	Reference
*H. sapiens*	*S. cerevisiae*		
ULK1	Atg1	Protein kinase in autophagy initiation	[11]
ATG2	Atg2	Lipid channel	[12,13]
ATG3	Atg3	E2 enzyme for lipidation of Atg8 family proteins	[14,15]
ATG4	Atg4	C-terminal proteolysis/delipidation of Atg8 family proteins	[14,16]
ATG5	Atg5	Ubiquitin-like protein, part of E3 complex	[14,17]
Beclin 1	Atg6/Vps30	PI3K subunit	[18,19]
ATG7	Atg7	E1 enzyme	[14,17]
LC3, GABARAP	Atg8	Ubiquitin-like protein in phagophore growth, autophagosome-lysome fusion	[20,14]
ATG9	Atg9	Lipid scramblase	[21,22]
ATG10	Atg10	E2 enzyme for Atg5-Atg12 conjugation	[14,17]
ATG12	Atg12	Ubiquitin-like protein, part of E3 complex	[14,17]
ATG13	Atg13	Part of ULK/Atg1 complex, scaffold	[23,11]
ATG14	Atg14	PI3K subunit, targeting to sites of phagophore formation	[18]
ATG16L1	Atg16	Part of E3 complex	[24,14]
CHMP4	Vps32	ESCRT-III subunit that mediates phagophore closure	[25,26]
DFCP1		PI3P binding protein on omegasomes	[27]
FIP200	Atg17	Part of ULK/Atg1 complex, scaffold	[23,11]
TSG101	Vps23	ESCRT-I subunit	[28]
VPS13	Vps13	Lipid channel	[29]
VPS15	Vps15	PI3K subunit	[18]
VPS34/PIK3C3	Vps34	Catalytic PI3K subunit	[18]
VPS37A	Vps37	ESCRT-I subunit, targets ESCRT machinery to holes in phagophores	[30,31]
WIPI1-4	Atg18	PI3P-binding protein involved in Atg2/ATG2 targeting	[32,33]

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
