# Peer review of "Autophagosome Biogenesis"

_cells, 2023, doi:10.3390/cells12040668_

Round 1
Reviewer 1 Report
The Review “Autophagosome biogenesis” by Yan Zhen and Harald Stenmark, it is a very intersting manuscript, well organised, fluently written and with captivating useful figures.
However, before publishing the review, the authors should:
11) insert at least two Tables with all molecules cited , indicating their main action/function, their role in yeast or in mammals, and also the main references, to allow the reader not only to have a summary of these but also to rapidly obtain the main references
22) In line 241, the PX-domain-containing protein SNX18 is cited: I suggest to expand this part to other SNXs, because in the current version it seems only another mention of the huge number of molecules involved
33) In Line 227: “by binding the ER-resident” there is an additional space
44) Taking into consideration how the final paragraph “CONCLUSIONS AND PERSPECTIVES” is organised , I suggest to change the title in “CONCLUSION: CHALLENGES AND PERSPECTIVES”
Author Response
We thank the three reviewers for insightful comments and excellent suggestions that have led to a further improvement of our manuscript. Please see the attachment.

Reviewer 2 Report
In the manuscript entitled " Autophagosome biogenesis”, Zhen and Stenmark present a review of the literature about the mechanisms of formation of autophagosomes in eukaryotes.
After a short introduction, the manuscript is organized in 11 short chapters and a conclusion/perspective. The first chapter presents the state of the art in the yeast, then a series of chapters describe the various organelles (RE, mitochondria, MAMs, plasma membrane, lipid droplets…) as sources of lipids in the formation of autophagosomes.The last chapter is dedicated to the closure of the phagosome and the role of ESCRT proteins.
The review is interesting, informative and relevant for the field. It covers a wide range of aspects about phagophore formation and many references are cited including recently published data.
Here are few suggestions that could improve the reading for the non-specialists of autophagy.
The question of how cargoes could influence the process of autophagosome biogenesis is not addressed (except for one sentence naming mitophagy). It is important because of the role of atg8 proteins in binding autophagy receptors an particularly in the context of selective autophagy.
Numerous proteins and molecular complexes are presented sometimes in details and sometimes rapidly, which is informative but makes the reading difficult for a non-specialist of autophagy. A table would be helpful. In such a table, indicating the specificities and differences between yeast and human proteins would be useful.
It is probable that the sites of formation and the origin of the lipids during autophagosome biogenesis are dependent of the cell types and the type of autophagy. The authors could have indicated the cellular context and the conditions of induction of autophagy used to obtain the various results. This could also be done in another table.
Remarks on the figures:
There is no illustration for the chapter 2, dedicated to the yeast, which make its reading quite difficult.
The figures 2 and 3 are quite simple and not very informative.
Figure 4 is nice but the starting point and the various steps will be easier to follow if authors simply add numbers in the figure, which they could refer to in the legend.
Author Response

(The authors gave the same response as above.)

Reviewer 3 Report
The review by Zhen and Stenmark is a well written, concise summary of the current knowledge on autophagosome biogenesis. The topic is timely and very suitable for the journal Cells. I have a few suggestions on how to update the text.
1. 1. Line 37 (page 1). Phagophores are classically described as cup-shaped double-membrane structures. It is misleading to state that phagophores are double-membrane structures. If phagophore is a double-membrane structure, then also the endoplasmic reticulum should be a double-membrane structure. My view is that phagophores are lined by a single membrane bilayer, similar to the ER. Once the phagophore closes and form an autophagosome, the autophagosome does have two membrane bilayers around the cargo. In other words, autophagosomes are double-membraned but phagophores are not.
2. 2. Also the classical cup-shape of phagophores, or at least phagophore precursors, has been recently challenged in Reference No 67 (Gudmundsson et al. 2022). This could be mentioned in the text.
3. 3. Line 46, page 1, ‘autophagosome of 400 nm’ -> autophagosome of 400 nm in diameter.
4. 4. Line 61, the PI3K complex. The yeast components are Vps15, Vps34, Atg 6 (not Vps6; Atg6 is also called Vps30), and Atg14 (not Vps14).
5. 5. Lines 132-133, ‘whereas others have 132 found that it is discontinuous [25]’. Reference No 25 was done using 2-dimensional electron microscopy, which does not allow conclusions on the continuity or discontinuity between two organelles. Please reword the sentence.
7. 6. The sentence on lines 318-321, starting ‘Studies from budding yeast…’ is confusing. Please rewrite.
8. 7. Lines 327-326 ‘interference with ATG2 function leads to accumulation of unclosed phagophores, presumably because lipid transfer from the ER is required for phagophore expansion [25,67].’ This statement should be changed. As stated above, Ref. No 25 was done using 2-dimensional EM that does not allow firm conclusions and phagophores or autophagosome being closed or not. The 3-dimensional EM in Ref. No 67, however, shows seemingly closed double-membraned ‘mini-autophagosomes’ in ATG2A and ATG2B double deficient cells.
Author Response

(The authors gave the same response as above.)
